# Synthesis, Structures and Electrochemical Properties of Lithium 1,3,5-Benzenetricarboxylate Complexes

**DOI:** 10.3390/polym11010126

**Published:** 2019-01-12

**Authors:** Pei-Chi Cheng, Bing-Han Li, Feng-Shuen Tseng, Po-Ching Liang, Chia-Her Lin, Wei-Ren Liu

**Affiliations:** 1Department of Chemistry, Chung-Yuan Christian University, Chungli 320, Taiwan; peegy430@hotmail.com (P.-C.C.); w6520325@hotmail.com (B.-H.L.); soon303304@hotmail.com (F.-S.T.); Rita7310@hotmail.com (P.-C.L.); 2Department of Chemical Engineering, Chung-Yuan Christian University, Chungli 320, Taiwan

**Keywords:** lithium, coordination polymers, electrochemical properties, framework structures

## Abstract

Four lithium coordination polymers, [Li_3_(BTC)(H_2_O)_6_] (**1**), [Li_3_(BTC)(H_2_O)_5_] (**2**), [Li_3_(BTC)(μ_2_-H_2_O)] (**3**), and [Li(H_2_BTC)(H_2_O)] (**4**) (H_3_BTC = 1,3,5-benzenetricarboxylatic acid), have been synthesized and characterized. All the structures have been determined using single crystal X-ray diffraction studies. Complexes **1** and **2** have two-dimensional (2-D) sheets, whereas complex **3** has three-dimensional (3-D) frameworks and complex **4** has one-dimensional (1-D) tubular chains. The crystal-to-crystal transformation was observed in **1**–**3** upon removal of water molecules, which accompanied the changes in structures and ligand bridging modes. Furthermore, the electrochemical properties of complexes **3** and **4** have been studied to evaluate these compounds as electrode materials in lithium ion batteries with the discharge capacities of 120 and 257 mAhg^−1^ in the first thirty cycles, respectively.

## 1. Introduction

Coordination polymers (CPs) [1,2,3,4,5,6,7] are promising precursors for material, consisting of metal ions linked together by organic bridging ligands. Due to their tunable crystalline structures and compositions, CPs have widely potential applications such as separation [8,9,10,11], adsorption [12,13,14], luminescence [15,16,17], catalysis [18,19,20,21,22], and drug delivery [23,24]. The synthesis of CPs can be achieved via employing metal ions as connected centers and varied organic ligands as linkers. Several different organic linkers have been widely applied, forming diverse structures and with emerging applications [25,26]. Reagents with multiple functional groups capable of reacting with metal ions play a key role in the synthesis of these coordination complexes. Among them, the 1,3,5-benzenetricarboxylic acid (H_3_BTC) has been frequently used in constructing metal CPs due to its multi-connectivity, i.e., from two to six carboxylate oxygen atoms can bridge metal ions to create 1-D chains, 2-D layers, and 3-D networks [27,28,29,30,31,32].

There have been few studies on lithium CPs with interesting electrochemical behaviors that could be used as electrodes in lithium ion batteries [33,34,35,36,37,38,39,40,41,42,43,44]. For the first time, two CPs of lithium terephthalate (Li_2_C_8_H_4_O_4_) and lithium *trans*–*trans*–muconate (Li_2_C_6_H_4_O_4_) displayed high electrode capacities up to 300 mAhg^−1^ [34]. These reported results show that those bonds between metal ions and organic ligands of lithium CPs may act as redox centers during electrochemical reactions. The interesting electrochemical results compare the commercial electrode materials and may have a lower capacity and coulombic efficiency. However, the abundant metal CPs have been intensely reported and may still have great potential function in various electrochemical applications. Recently, we reported that lithium CPs were used as negative electrode material which showed a capacity of approximately 100 mAhg^−1^ [31,32]. Continuing this interesting research, the structural analysis and electrochemical properties for the new lithium CPs are presented. In this study, we report the synthesis, structures and electrochemical properties of four new lithium CPs, [Li_3_(BTC)(H_2_O)_6_] (**1**), [Li_3_(BTC)(H_2_O)_5_] (**2**), [Li_3_(BTC)(μ_2_-H_2_O)] (**3**), and [Li(H_2_BTC)(H_2_O)] (**4**).

## 2. Experimental Section

### 2.1. Materials and General Methods

The metal ion sources, organic ligands, solvents and reagents were received and used from possible commercial agents. The reactions were processed by heating the reaction mixtures in Teflon-lined digestion bombs (volume with 23 mL) to the design temperatures under autogenous pressure and followed by slow decreasing at 6 °C/h to room temperature. All the elemental analysis measurements were performed with crystal samples (~5 mg, each sample) to analyze the organic compositions of the lithium CPs. Infrared (IR) spectra were measured (400–4000 cm^−1^ region) with JASCO FT/IR-460 spectrophotometer (JASCO, Easton, MD, USA) by using KBr disks. Thermal gravimetric analyses (TGA) by DuPont TA Q50 were measured as powder samples under flowing N_2_ with a temperature increasing rate of 10 °C/min.

### 2.2. Synthesis of [Li_3_(BTC)(H_2_O)_6_] (***1***)

1,3,5-Benzenetricarboxylic acid (H_3_BTC, 0.210 g, 1.0 mmol) was placed in a 20 mL sample bottle containing LiOH (0.0719 g, 3.0 mmol) and 6.0 mL of THF (99%) was added. The mixture was then stirred for 3 h to afford a white solid. The precipitate was filtered and dried under a vacuum to give a white product with some colorless crystals. Yield: 0.3109 g (92.52% based on Li). Elemental analysis found/calcd.: C, 32.85/32.17; H, 3.96/4.50% for **1**. IR (KBr, cm^−1^): 3415(br), 1620(s), 1566(s), 1440(s), 1373(s), 1106(m), 762(m), 730(m), 688(m), 598(w), 561(m).

### 2.3. Synthesis of [Li_3_(BTC)(H_2_O)_5_] (***2***)

1,3,5-Benzenetricarboxylic acid (H_3_BTC, 0.210 g, 1.0 mmol) was placed in a 20 mL sample bottle containing LiOH (0.0719 g, 3.0 mmol) and MeOH/H_2_O (5.0 mL/1.0 mL) was added. The mixture was then stirred for 3 h to afford a clear solution and diethyl ether was added to induce colorless crystals. Colorless needle crystals were obtained by slow diffusion of diethyl ether into MeOH solution of the compound for several days. Yield: 0.2341 g (73.61% based on Li). Elemental analysis found/calcd.: C, 34.61/33.99; H, 3.82/4.12% for **2**. IR (KBr, cm^−1^): 3531(br), 3375(m), 3363(m), 1642(s), 1616(s), 1565(s), 1442(s), 1393(s), 1374(s), 1109(br), 937(br), 774(s), 772(s), 692(w).

### 2.4. Synthesis of [Li_3_(BTC)(μ_2_-H_2_O)] (***3***)

A mixture of LiOH (0.0719 g, 3.0 mmol), 1,3,5-benzenetricarboxylic acid (H_3_BTC, 0.210 g, 1.0 mmol), alcohol/water (5.0 mL/1.0 mL) mixture (MeOH, substitute for EtOH or IPA was used) were placed in a 23 mL Teflon-lined stainless container, which was sealed and heated at 120 °C for 2 days under autogenous pressure and then cooled slowly at room temperature. The colorless block crystals were collected with yield of 0.0806 g, 0.1772 g and 0.1889 g (32.77%, 72.04% and 76.8% based on lithium reagent). Elemental analysis found/calcd.: C, 43.98/43.95; H, 2.26/2.05% for **3**. IR (KBr, cm^−1^): 3414(m), 3281(m), 1627(s), 1570(s), 1441(s), 1388(s), 1106(w), 936(w), 768(w), 727(w).

### 2.5. Synthesis of [Li(H_2_BTC)(H_2_O)] (***4***)

A mixture of LiOH (0.0240 g, 1.0 mmol), 1,3,5-benzenetricarboxylic acid (H_3_BTC, 0.210 g, 1.0 mmol), H_2_O (1.0 mL), and MeCN (5.0 mL) were placed in a 23 mL Teflon-lined stainless container, which was sealed and heated at 120 °C for 2 days under autogenous pressure and then cooled slowly at room temperature. The colorless block crystals were collected with yield of 0.2024 g (86.46% based on lithium reagent). Elemental analysis found/calcd.: C, 46.13/46.18; H, 3.16/3.01% for **4**. IR (KBr, cm^−1^): 3259(br), 1706(m), 1617(m), 1573(m), 1326(m), 1246(s), 1111(m), 910(m), 752(s), 696(s), 604(s). 

### 2.6. Single-Crystal Structure Analysis

The X-ray diffraction measurements were processed by Bruker AXS SMART APEX II diffractometer (Bruker AXS, Madison, WI, USA) (Mo-Kα radiation, graphite monochromator, λ = 0.71073 Å). The raw data were collected with a scan by combination of both ω and φ angle. Data analysis was first corrected for Lorentz and polarization methods, and the program *SADABS* in *APEX2* [45] was applied to run the absorption correction. Further calculations and refinements were performed by using *APEX2* programs. The crystallographic data are summarized in Table 1. The metal to oxygen bond distances are listed in Appendix A. Other details are given in the Appendix A.

### 2.7. Electrochemical Measurements

The electrochemical measurements of CPs **3** and **4** were achieved by thoroughly manually mixing the particles with a 30 wt % amount of SP-type carbon black and 10 wt % binder of polyvinylidene difluoride (PVDF, G-580, TCI). The CPs and SP conductor materials were well-mixed first, and then the binder of PVDF in *N*-methylpyrrolidinone (NMP) was added and mixed again. The final slurry was transferred and coated onto copper foil (10 μm, Nippon Foil Co, Nippon Foil Mfg. Co, Yodogawa-Ku, Osaka, Japan). The dried electrode film was compressed by a roller at room temperature to make the film smoother and more compact. After drying at 100 °C for 6 h under vacuum, the circular electrode disks with area about 1.37 cm^2^ were punched out of the larger coated foil sheets and weighed. The mixture was rolled into thin sheets and pressed into 7 mm circular disks in diameter as electrodes. The typical electrode mass and thickness were 8 to 13 mg and 0.03 to 0.08 mm. The electrochemical measurements were recorded in a standard 2032-coin type sealed in an Ar-filled glove box. A lithium metal foil was then used as the counter and reference electrodes. One piece of Celgard 2320 microporous membrane separator (Celgard, New Jersey, NJ, USA) was thoroughly soaked with 1 M LiPF_6_ EC/DMC 1:1 electrolyte (Merck). The electrode cycling tests were recorded by a battery testing system on a Maccor Model-2200 (Maccor, Tulsa, OK, USA). Cyclic voltammetry (CV) was determined on Autolab PGSTAT-101 (Metrohm UK Ltd, Runcorn, UK) station in combination with a powerful Nova software electrochemical analyzer (Metrohm UK Ltd, Runcorn, UK) at 1 mVs^−1^ in the three-electrode system with 1 M solution of LiPF_6_ containing EC:DMC = 1:1 by weight.

## 3. Results and Discussion

### 3.1. Structural Description of [Li_3_(BTC)(H_2_O)_6_] (***1***)

The crystal structure of **1** shows a 2-D layered packing. The asymmetric unit contains three Li atoms, one BTC ligand and six coordinated water molecules. The Li1 atom is coordinated by one carboxylate oxygen atom from BTC ligand and three water molecules (1.856(4)–1.991(3) Å); whereas the Li2 and Li3 atoms are coordinated by two oxygen atoms from two BTC ligands and two water molecules (Li2:1.947(4)–2.111(4) Å, Li3:1.905(3)–1.988(3) Å) (Figure 1a). The BTC ligand is coordinated to five lithium atoms with a μ_5_ environment (Figure 1b). Figure 1c clearly shows that Li atoms are linked by the BTC ligands generating the 2-D sheets. The hydrogen bonds between the layers by oxygen atoms of coordinated water and the carboxylate group were observed in the crystal structure of **1** (Figure 1d).

### 3.2. Structural Description of [Li_3_(BTC)(H_2_O)_5_] (***2***)

The crystal structure of **2** also shows a 2-D layered packing along the *ab* plane. The basic structural unit consists of three Li atoms, one BTC ligand and five coordinated water molecules. The lithium atoms are four coordinated by two oxygen atoms of two BTC ligands and two oxygen atoms of coordinated water molecules, with typical Li-O bond lengths ranging from 1.869(2) to 2.091(3) Å (Figure 2a). The BTC anion bonds to six lithium atoms and adopts a μ_6_ environment (Figure 2b).

### 3.3. Structural Description of [Li_3_(BTC)(μ_2_-H_2_O)] (***3***)

The crystal structure of **3** shows a 3-D connection in which the asymmetric unit contains three Li atoms, one BTC ligand, and one μ_2_-H_2_O molecule. Among the three Li atoms in the asymmetric unit, the Li1 atom is coordinated to four O atoms of four BTC ligands forming a distorted LiO_4_ tetrahedral geometry, whereas, the other Li atoms, Li2 and Li3, are four coordinated and forming a distorted tetrahedral geometry which consists with three carboxylate O atoms from three BTC ligands and a μ_2_-aqua bridge between the lithium atoms forms (Figure 3a). The BTC ligand is coordinated to ten lithium atoms with a μ_10_ environment (Figure 3b). Appendix A shows that all the bond length of Li-O bonds of **1** range from 1.886(2) to 2.069(2) Å. As shown in Figure 3c, all the distorted tetrahedral LiO_4_ are edged-sharing and create 1-D inorganic motifs along the [101] direction. These chains are further linked together by the BTC ligands along the *c*-axis and generate the 3-D framework.

### 3.4. Structural Description of [Li(H_2_BTC)(H_2_O)] (***4***)

The crystal structure of **3** shows 1-D chains in which the asymmetric unit contains half Li atom, half H_2_BTC ligand, and a half-coordinated water molecule. The Li ion is coordinated by two oxygen atoms of carboxylate groups from two H_2_BTC ligands, and two oxygen atoms of the coordinated water molecule (Figure 4a). The center of symmetry at the center of the H_2_BTC ligand has been observed in **4**. It is noteworthy that in Figure 4b the H_2_BTC ligand has been shown to be coordinated to two lithium atoms using its two carboxylate groups and adopts a μ_2_ environment. As shown in Figure 4c, the LiO_4_ distorted tetrahedral share corners and create inorganic motifs in the form of 1-D chains running along the *a*-axis. These chains are linked together by μ_2_ links of the BTC ligands along the *bc*-plane to generate the 1-D tubular chains of **4**. Moreover, these chains are connected to each other through hydrogen bonds between coordinated water and carboxylate oxygen atoms (Figure 4d).

### 3.5. Thermogravimetry Analysis

The thermogravimetry analysis (TGA) measurements have been studied for the understanding of thermal stability. As shown in Figure 5, the TGA curves of the of **1** and **2** have weight loss in the first step of 31.67% and 27.93% before 300 °C which may correspond to the loss of coordinated water molecules; and the further weight loss of 57.89% and 56.76% has been observed at 500–800 °C. No obvious weight loss for **3** was displayed before 250 °C, indicating the μ_2_-H_2_O molecules are strongly bonding to the lithium centers. For **3**, the first weight loss of 7.70% was observed at 250–280 °C which may correspond to the loss of coordinated water molecules (calcd.: 7.32%), indicating that the dehydration compound could be stable at about 500 °C. The second weight loss of 70.22% has been observed at the temperature range of 500 to 800 °C. The complex **4** has no weight change up to 120 °C and displays the first weight loss of 8.00% (calcd.: 7.70%) at about 120–150 °C, corresponding to the partial loss of coordinated water molecules. Then, the second weight loss of 82.92% was observed at a 250–800 °C temperature range. 

### 3.6. Crystal-To-Crystal Transformation

The water molecule coordinated complexes **1**–**3** show interesting structural transformations when the water molecules are removed. To confirm the transformation in **1**–**3**, we first checked their pristine structures by measuring the PXRD patterns. Appendix A show that the powder patterns of these three complexes are consistent with those calculated from single-crystal X-ray data. Exploration of the possible structural changes associated with this transition by PXRD pattern revealed a phase transformation occurring when the **1** was heated above 50 °C, the pattern became similar to complex **2** (Appendix A). We then measured the PXRD patterns of **1** and **2**, Appendix A, showing that the patterns of **1** and **2** change at 100 °C and both have similar PXRD patterns to that for complex **3**, indicating the crystal-to-crystal transformation from **1** to **3** and **2** to **3**. At 300 °C, the sample is dehydrated and the powder pattern is retained without losing crystallinity. Notably, when the dehydrated sample was exposed to H_2_O vapor, that is, the dry sample was placed in a glass desiccator beside a beaker filled with H_2_O, it reabsorbed the lost H_2_O. The PXRD pattern of the rehydrated species (Appendix A) is almost the same as that of the freshly synthesized material, suggesting that the original crystalline phase of **3** was regenerated. The transformation from dehydrated complex to **3** is thus reversible. However, attempts to rehydrate **3** to obtain **1** or **2** have not been successful. The dehydration process from **1** to **2**, **1** to **3** and **2** to **3** is therefore irreversible. The first step transformation from phase **1** to **2**, which happens at 50 °C, involves the change of coordination water six to five, μ_5_ to μ_6_ coordination mode. The second step transformation from phase **2** to **3**, maintain one coordination water, and μ_10_ coordination mode of BTC at 100 °C.

### 3.7. Electrochemical Properties

Figure 6 shows the oxidation/reduction cyclic voltammogram of the complexes **3** and **4** at the scan rate of 1 mVs^−1^ for the first five cycles. Notably, the CV curve of complexes **3** and **4** in the first cycles were different from the CV curves of the subsequent cycles. For the first cycle, it illustrated a broad and irreversible reduction peak ranging from 1.6 to 0.01 V and 1.5 to 0.01 V, respectively. This irreversible capacity loss in the first cycle can be attributed to the formation of solid electrolyte interface (SEI) on the surface of the active electrode. In the case of the oxidation processed, complexes **3** and **4** displayed two obvious peaks located at 0.43 and 1.0 V, which might result from the formation and breaking of the Li-O bond during the charge process.

Figure 7 shows the cycling tests of complexes **3** and **4** at 0.1 C with 30 cycles. The chare/discharge capacities of complex **3** in the first cycle was found to be 402 and 119 mAhg^−1^, respectively. The coulombic efficiency was determined to be 30% in Figure 8. After 30 cycles, **3** showed a reversible capacity of ~120 mAhg^−1^ with excellent stability. In the case of complex **4**, the charge/discharge capacity in the first cycle was 1276 and 250 mAhg^−1^, respectively. It was to be noted that the cyclability of complex **4** was very much comparable with that of complex **3**. In addition, the reversible capacity of complex **4** was maintained as 257 mAhg^−1^ even after 30 cycles without decay. After the first cycle, the stable SEI film of complex **3** and complex **4** exhibited excellent structural stability during charge and discharge processes. Then, after 30 cycles, the reversible capacity of complex **4** (257 mAhg^−1^) was even better than that of complex **3**. During the first cycle, the rapid decrease in charge/ discharge capacities of complex **3** and complex **4** might be due to the irreversible electrochemical nature of the lithium complexes, which is quite common for lithium-containing compounds. These results reveal that both lithium coordination polymers have active electrochemical properties and display similar capacity, in comparison to previously reported studies where they ranged from about 100 to 600 mAhg^−1^ [33,34,35,36,37,38,39,40,41,42,43,44].

The rate capability tests of complex **3** and complex **4** are presented in Figure 9. The results show the ability of the material to achieve excellent specific storage capacities at different *C*-rates. The C rate is defined as the current to charge or discharge the nominal capacity in 1 h. Thus, 1C means the current we use to charge or discharge our battery by 1 h. In this study, 1C = 400 mA/g. The reversible capacities were approximately 119, 99, 83 and 73 mAhg^−1^ for complex **3**, 265, 212, 154 and 112 mAhg^−1^ for complex **4** at 0.1C, 0.2C, 0.5C and 1C respectively.

To further investigate the impedance in the interfaces, these materials were studied by impedance spectroscopy shown in Figure 10a,b. The AC impedance was measured after the cells were galvanostatically charged (0.1 C) to 0.1 V. In the complex impedance plot, the semicircle has a high-frequency intercept of the curve with the impedance axis that corresponds to the ionic resistance of the electrolyte (*R_e_*) caused by the migration of the Li ions in the electrolyte. At the low-frequency intercept of the semicircle with the *Z*’ axis, the resistance related to the charge transfer resistance between the electrolyte and the active material (*R_et_*) can be identified [40,41,42]. The low-frequency tail represents the third region with typical Warburg behavior, which is related to the diffusion of lithium ions in the cathode material. At the high-frequency region, the related resistance was attributed to both inter-particle electronic contacts and ionic migration through the passivation layer, the polymer layer, and the conductive additives. As shown in Figure 10, the semicircle resistance at the high-frequency region of the reported electrode, complex **4** were lower than that of **3**. While the diffused magnitudes of the lithium ions were similar.

The lithium-ion diffusion coefficient can be calculated using the following equation [43,44]:(1)$$D=\frac{R^{2}T^{2}}{2A^{2}n^{4}F^{4}C^{2}\sigma_{w}{}^{2}}$$
where *R* is the gas constant (8.314 J·K^−1^·mol^−1^), *T* is the absolute temperature (293.15 K), *A* is the surface area of the electrode (~1.54 cm^2^), *n* is the number of electrons per molecule during oxidization (*n* = 3), *F* is Faraday’s constant (96500 C·mol^−1^), *σ***_w_** is Warburg impedance coefficient and *C* is the concentration of lithium ions (0.001 mol cm^−3^) in electrolyte. The *σ*_w_ values are obtain from the slope of Bode plot. Based on equation (1), diffusion coefficients of lithium in complex **3** and complex **4** were calculated to be 7.6 × 10^−16^ cm^2^/s and 3.0 × 10^−15^ cm^2^/s, respectively. Apparently, Li^+^ diffusion coefficient of complex **4** was much higher than that of complex **3**. The result was inconsistent with rate capability tests data shown in Figure 8.

## 4. Conclusions

In the present study, we have successfully synthesized three new lithium CPs using 1,3,5-Benzenetricarboxylic acid (H_3_BTC) and lithium(I) ions. The coordination variability of H_3_BTC is even richer with polycarboxylate anions used as linkers with bridging modes *μ*_n_, where *n* varies between 2, 5, 6 and 10. The various inorganic motifs were constructed using H_3_BTC ligands forming 1-D, 2-D and 3-D structures. The crystal-to-crystal transformation involves not only removing the coordination water molecules but also the breaking and formation of coordination bonds. In particular, the two-step crystal-to-crystal transformation causes significant crystallographic change that involves coordination mode of BTC ligand. The Li_3_BTC based complexes exhibit unusual structural diversity triggered by water molecules only owing to the multi-connectivity of ligands. In addition, the investigation of lithium CPs based on carboxylates represents an interesting and fruitful part of solid-state chemistry and materials chemistry. Moreover, the preliminary electrochemical studies of newly synthesized lithium CPs have supported the potential application in electrochemical chemistry. Further investigation based on the lithium metal with aromatic carboxylic acid ligand is still in progress.

## Figures and Tables

**Figure 1 polymers-11-00126-f001:**
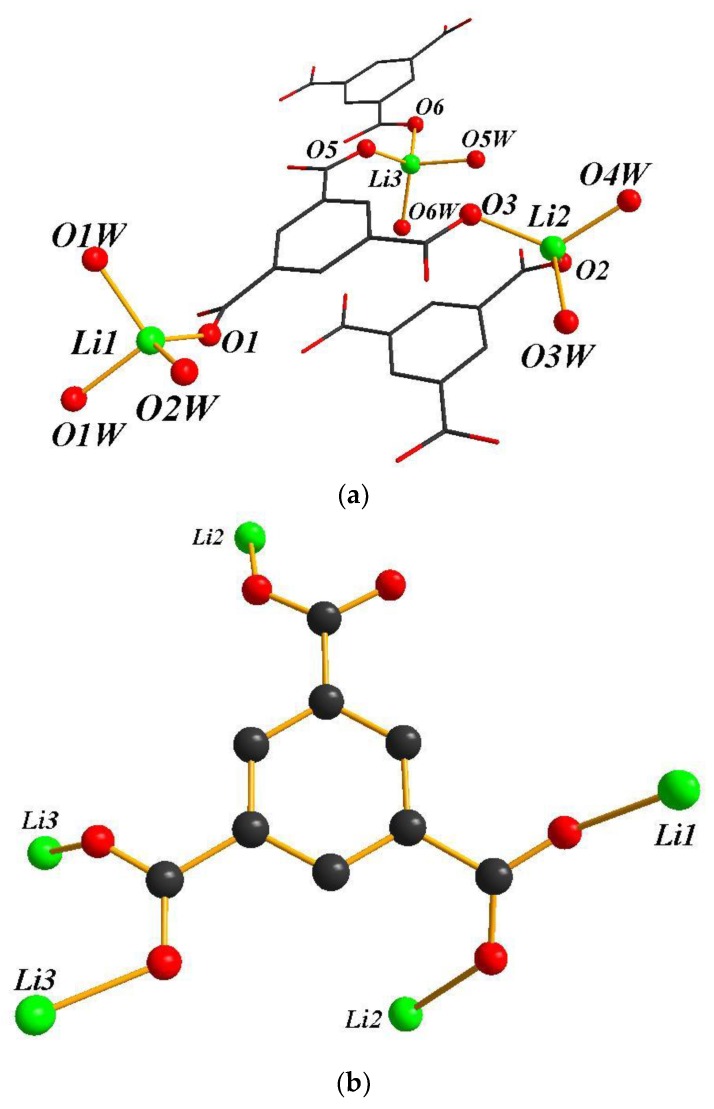
(**a**) The coordination spheres of lithium atoms in **1**; (**b**) the coordination model of the BTC ligand in **1**; (**c**) the one-layer structure view of **1** (H atoms were omitted for clarity). (**d**) The 2-D sheets viewed along the *b*-axis of **1**.

**Figure 2 polymers-11-00126-f002:**
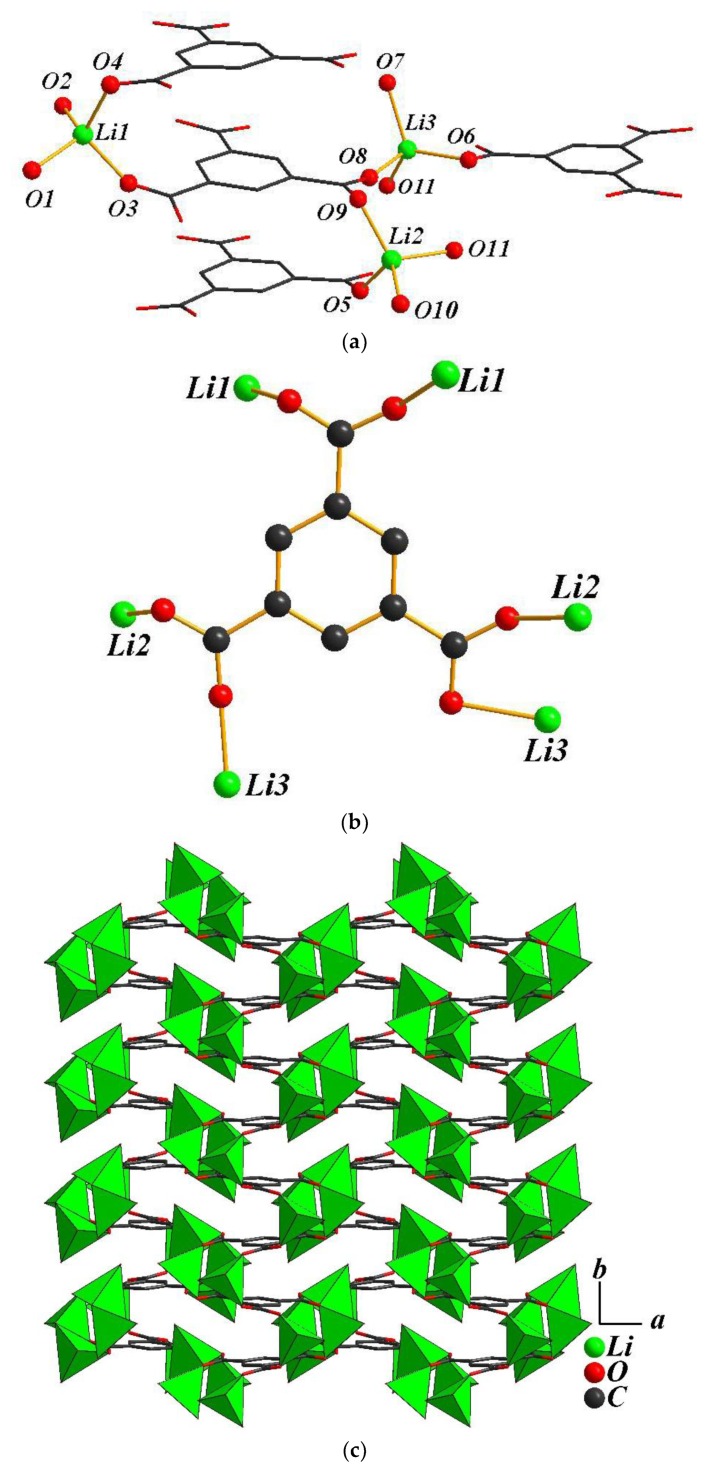
(**a**) The coordination spheres of lithium atoms in **2**; (**b**) the coordination model of the BTC ligand in **2**; (**c**) the one-layer structure view along the *c*-axis of **2** (H atoms were omitted for clarity); (**d**) the 2-D sheets viewed along the *b*-axis of **2**.

**Figure 3 polymers-11-00126-f003:**
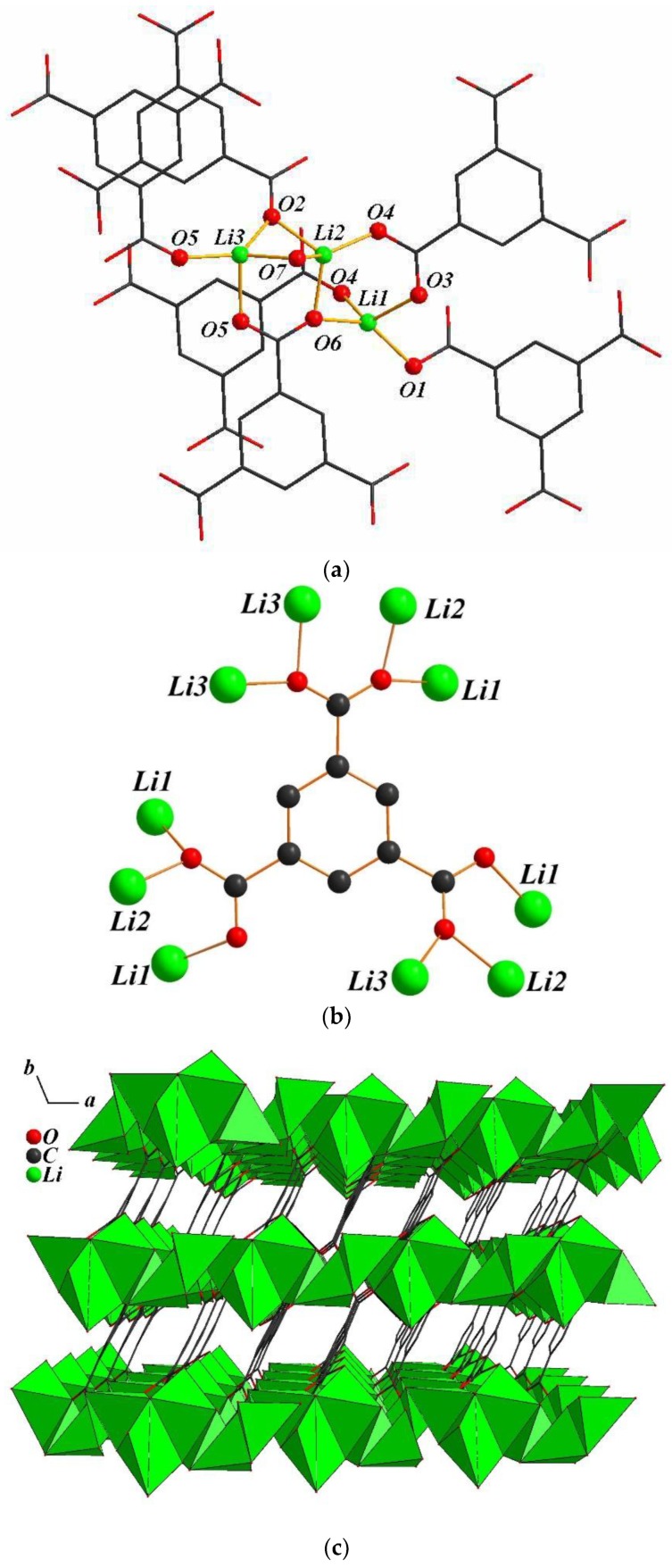
(**a**) The coordination spheres of lithium atoms in **3**; (**b**) the coordination model of the BTC ligand in **3**; (**c**) the 3-D network viewed along the *c*-axis with edge-sharing 1-D chains in compound **3** (H atoms were omitted for clarity).

**Figure 4 polymers-11-00126-f004:**
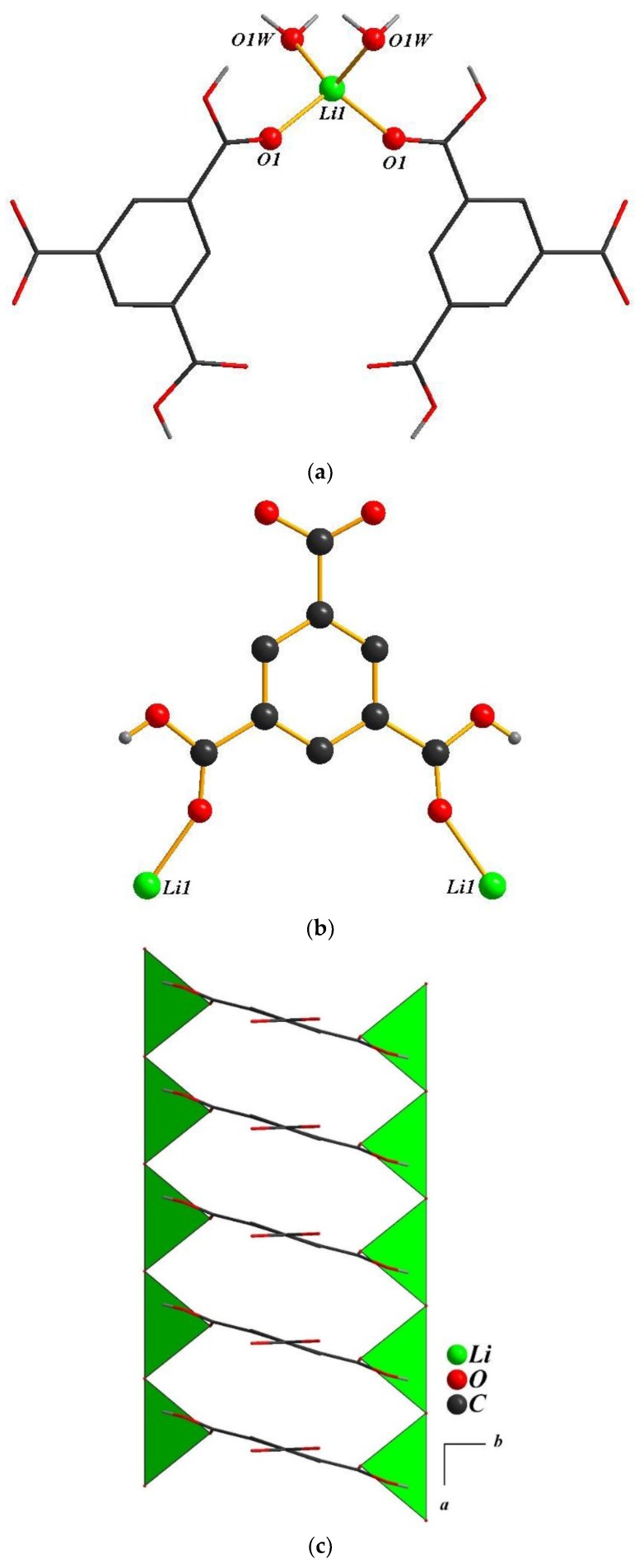
(**a**) The coordination spheres of lithium atoms in **4**; (**b**) the coordination model of the BTC ligand in **4**; (**c**) the 1-D chain of **4**. (**d**) The 1-D chains viewed along the *a*-axis of **4**.

**Figure 5 polymers-11-00126-f005:**
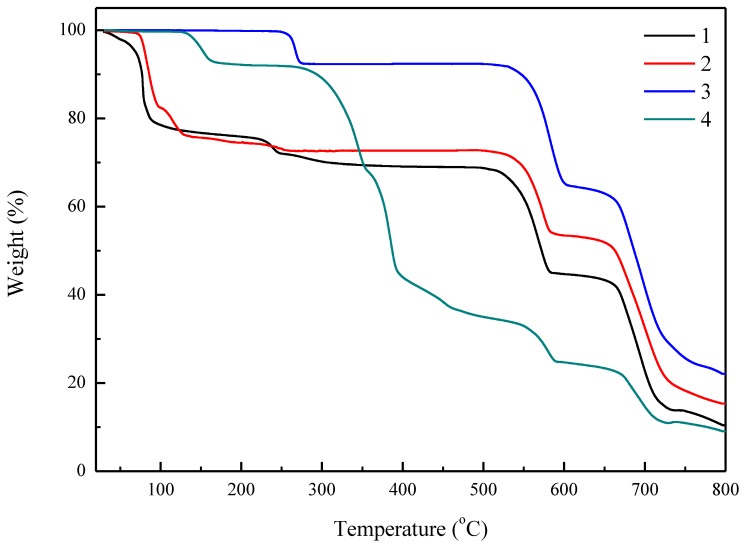
TGA curves for the compound **1**–**4** with a heating rate of 10 °C/min.

**Figure 6 polymers-11-00126-f006:**
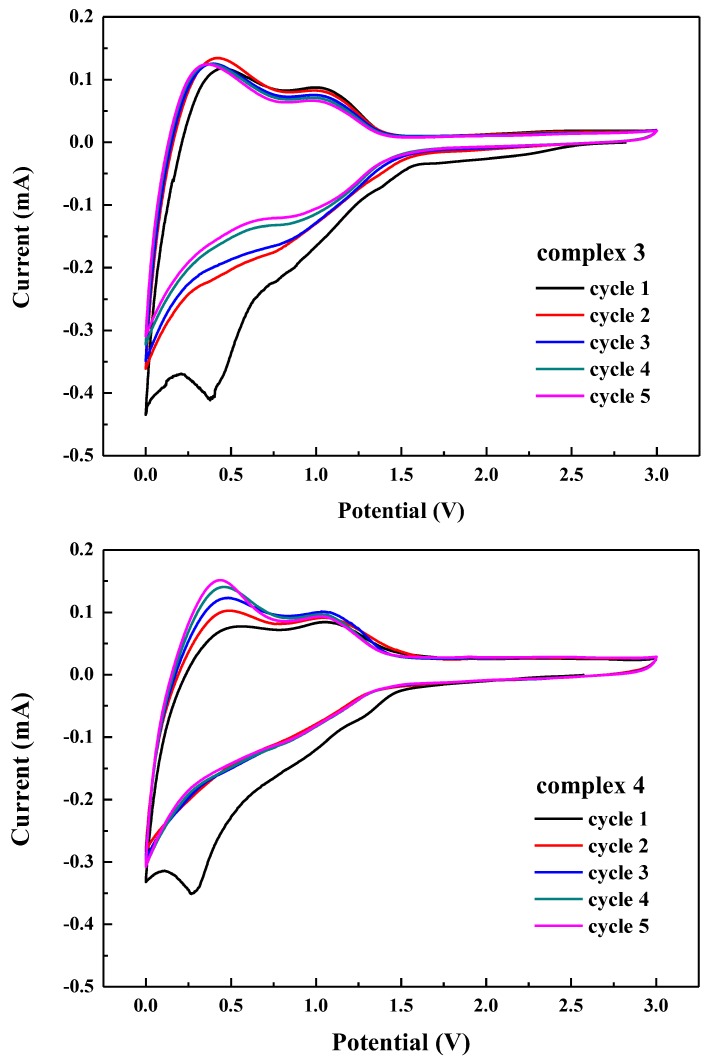
Cyclic voltammograms of complexes **3** and **4** in 1M LiPF_6_ EC/DMC (1:1 *V*/*V*), which was recorded in the first five cycles at a scanning rate of 1 mVs^−1^.

**Figure 7 polymers-11-00126-f007:**
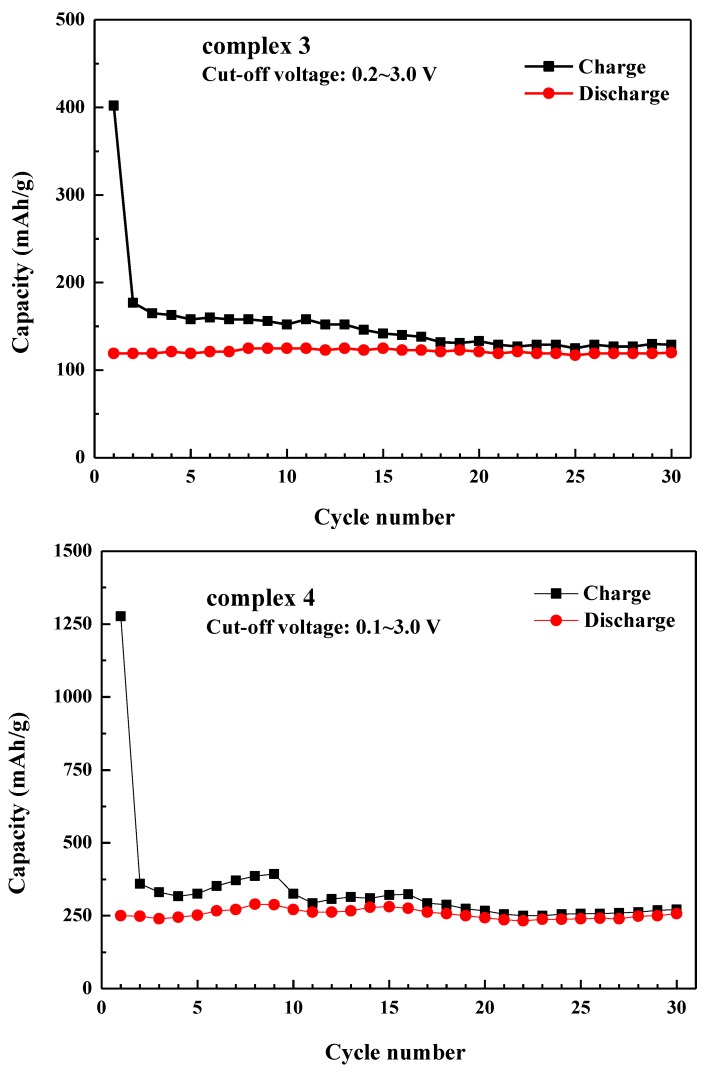
The Galvanostatic cycling of complexes **3** and **4**. Conditions: cycling rate, C/10; potential limits, 3.0 and 0.1 V; electrolyte, 1M LiPF_6_ in EC:DMC (1:1 *V*/*V*); 60% active material; 30% Super P; 10% PVDF binder.

**Figure 8 polymers-11-00126-f008:**
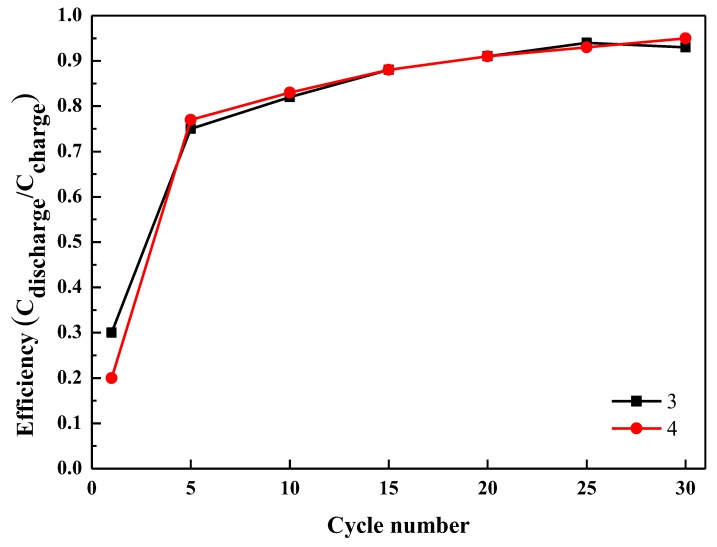
The coulombic efficiency vs. cycle number of the complexes **3** and **4** organic electrode.

**Figure 9 polymers-11-00126-f009:**
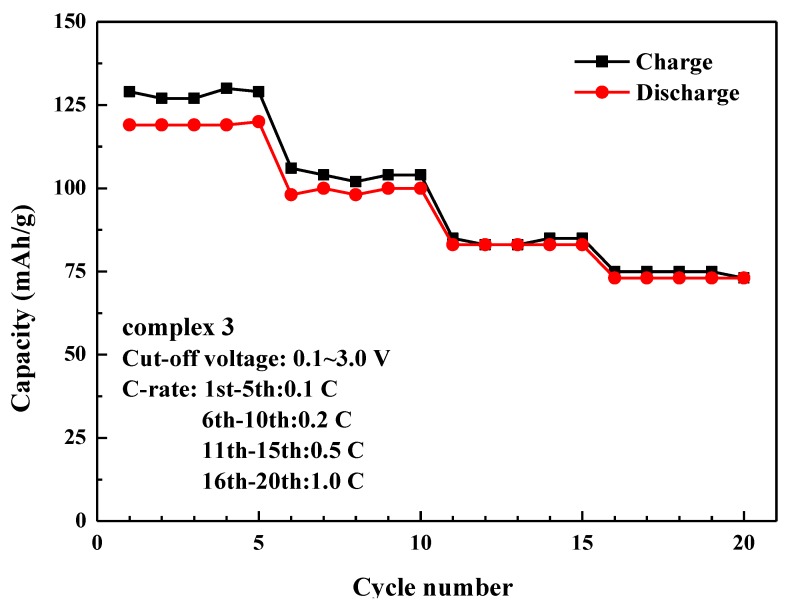
Rate capability tests of complexes **3** and **4** with potential window 0.1–3.0 V.

**Figure 10 polymers-11-00126-f010:**
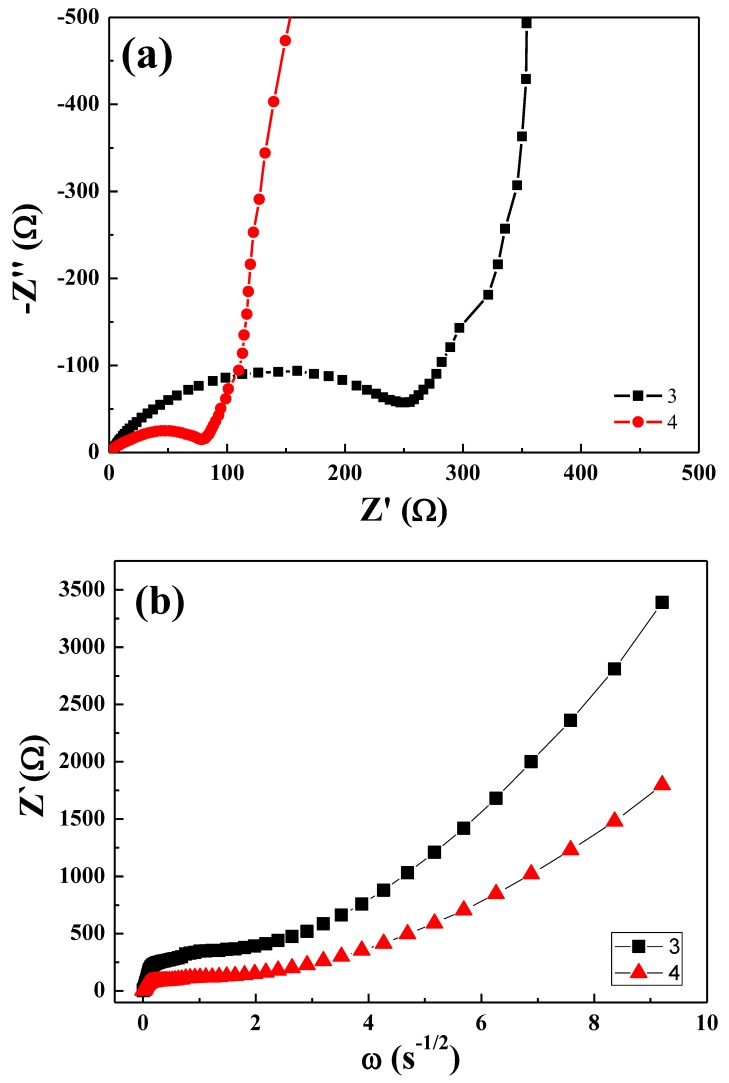
(**a**) The AC impedance of complex **3** and complex **4**; (**b**) diagram of Z’ and ω^−1/2^. The slope of diagram is Warburg impedance coefficient.

**Table 1 polymers-11-00126-t001:** Crystallographic data for **1**–**4**.

Compound	1	2	3	4
Formula	C_9_H_15_Li_3_O_12_	C_9_H_13_Li_3_O_11_	C_9_H_5_Li_3_O_7_	C_9_H_7_LiO_7_
Formula weight	336.03	318.01	245.95	234.09
Crystal habit	Lamellar	Acicular	Equant	Acicular
Crystal system	Triclinic	Orthorhombic	Triclinic	Orthorhombic
Space group	*P* ī	*Pbca*	*P* ī	*Pnnm*
*a*(Å)	6.9879(5)	13.460(3)	7.4099(2)	3.5451(2)
*b*(Å)	10.9331(8)	7.124(2)	8.0589(2)	13.4375(9)
*c*(Å)	11.2513(8)	28.197(6)	9.4320(3)	20.3671(13)
α(°)	61.102(4)	90.00(3)	110.626(2)	90
β(°)	72.371(4)	90.00(3)	104.348(2)	90
γ(°)	82.509(3)	90.00(3)	105.272(1)	90
Volume(Å^3^)	717.08(9)	2703.8(11)	470.95(2)	970.23(11)
*Z*	2	8	2	4
*D*_calc_(gcm^−3^)	1.556	1.562	1.734	1.603
*μ*(mm^−1^)	0.142	0.141	0.145	0.139
Collection *T* (K)	296(2)	296(2)	295(2)	295(2)
*λ*(Å)	0.71073	0.71073	0.71073	0.71073
Reflections collected	12523	24666	19445	8468
Independent reflections	3503	3362	4354	1243
R(int)	0.0338	0.0519	0.0486	0.0238
Goodness-of-fit on F^2^	1.041	1.047	1.063	1.033
*R*_1_ [*I* > 2*σ*(*I*)]	0.0442	0.0358	0.0375	0.0351
*wR*_2_ [*I* > 2*σ*(*I*)]	0.1042	0.0885	0.1101	0.0995
*R*_1_ [all data]	0.0670	0.0507	0.0508	0.0420
*wR*_2_ [all data]	0.1140	0.0964	0.1301	0.1052

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
