# Peer review of "Synthesis, Structures and Electrochemical Properties of Lithium 1,3,5-Benzenetricarboxylate Complexes"

_polymers, 2019, doi:10.3390/polym11010126_

Round 1

Reviewer 1 Report

The work reported new lithium coordination polymers for lithium ion batteries. The following issues should be carefully addressed.

1. In the introduction part, the advantages of using lithium coordination polymers should be clarified. It should be noted that such a low capacity and low coulombic efficiency of this kind materials may be not popular for lithium ion batteries.

2. How to define 1C in the present study? This should be given in the maintext.

3. It is better to compare the present electrochemical properties with the previously reported studies.

4. The EIS anlysis of the Nyquist plot is confusing, especially for the charge transfer resistance. Please refer to and cite the references of ACS Appl. Mater. Interfaces 2018, 10, 18816–18823, Chem. Eng. J. 2018, 354, 220-227, J. Mater. Chem. A 2018, 6, 13419-13427, for a better readership. The sentence of As shown in Fig. 10,... in lines 230-232 should be rearranged. The term of semicirle resistance is not a professional name, which should be charge transfer resistance. It is better to present the value of charge transfer resitance (semicircle).

5. The description of Eq (1) should also include some literature to solid it, such as Chem. Eng. J. 2017, 328, 591–598, and Energy Storage Mater. 2018, 12, 94-102. In addition, how to obtain the σw values should be given.

6. There are some gramar mistakes, such as in line 244, shows should be show. Please double check the manuscript.

Author Response

The work reported new lithium coordination polymers for lithium ion batteries. The following issues should be carefully addressed.

1. In the introduction part, the advantages of using lithium coordination polymers should be clarified. It should be noted that such a low capacity and low coulombic efficiency of this kind materials may be not popular for lithium ion batteries.

Response: We thank the reviewer for the careful reading and helpful comments on introduction section. These new materials of lithium coordination polymers have low capacity and low coulombic efficiency was mentioned and update in the introduction section. More research motivation and intention were added into the introduction section.

2. How to define 1C in the present study? This should be given in the main text.

Response: The C rate is defined as the current to charge or discharge the nominal capacity in 1 hour. Thus, 1C means the current we use to charge or discharge our battery by 1 hour. The definition of 1C will be added in the revised manuscript. 

3. It is better to compare the present electrochemical properties with the previously reported studies.

Response: The present electrochemical properties were compared with previously reported studies and updated in the main-text.

4. The EIS anlysis of the Nyquist plot is confusing, especially for the charge transfer resistance. Please refer to and cite the references of ACS Appl. Mater. Interfaces 2018, 10, 18816–18823, Chem. Eng. J. 2018, 354, 220-227, J. Mater. Chem. A 2018, 6, 13419-13427, for a better readership. The sentence of “As shown in Fig. 10,...” in lines 230-232 should be rearranged. The term of “semicirle resistance” is not a professional name, which should be charge transfer resistance. It is better to present the value of charge transfer resitance (semicircle).

Response: Thank you for reviewer’s comments. The corresponding discussion of EIS analysis will be refined, also including the charge transfer resistance, in the revised manuscript. Of course, these importance references will be cited in the revised manuscript. These mistakes and typo errors all will be corrected in the revised manuscript.

9(h) Wang, J.; Liu, H.; Liu, H.; Hua W.; Shao, M. ACS Appl. Mater. Interfaces, 2018, 10, 18816.

9(i) Zhang, W.; Li, J.; Zhang, J.; Sheng, J.; He, T.; Tian, M.; Zhao, Y.; Xie, C.; Mai, L.; Mu, S. Chem. Eng. J. 2018, 354, 220.

9(j) Sun, H.; Wang, J.; Zhang, H.; Hua, W.; Li, Y.; Liu, H.; H. J. Mater. Chem. A 2018, 6, 13419.

5. The description of Eq (1) should also include some literature to solid it, such as Chem. Eng. J. 2017, 328, 591–598, and Energy Storage Mater. 2018, 12, 94-102. In addition, how to obtain the σw values should be given.

Response: Thank you for reviewer’s comments. We will add these references to solid the Eq (1) in the revised manuscript. The σw values are obtain from the slope of Bode plot. Detail information will be given in the revised manuscript.

9(k) Wang, J.; Liu, H.; Liu, H.; Fu, Z.; Nan, D.; Chem. Eng. J. 2017, 328, 591.

9(l) Sheng, L.; Liang, S.; Wei, T.; Chang, J.; Jiang, Z.; Zhang, L.; Zhou, Q.; Zhou, J.; Jiang, L.; Fan, Z., Energy Storage Mater. 2018, 12, 94.

6. There are some grammar mistakes, such as in line 244, shows should be “show”. Please double check the manuscript.

Response: The English and grammar in this manuscript was improved and checked by native English-speaking colleague. All changes were marked with red color.

Reviewer 2 Report

Comments:

This manuscript presents syntheses, solid-state structures, and electrochemical properties of lithium 1,3,5-benzenetricarboxylate (BTC) salts. Although this is a much simplest study based on Li(I)–BTC salts, the technical works are beautiful and careful. In addition, these salts, especially compounds 3 and 4, may potentially find use in applications such as Li ion batteries. Therefore, this manuscript is recommended for publication in Polymers. I have only three minor comments as below:

(1) In the experimental section, please report the characteristic bands of IR spectrum for compound 2, which have been studied.

(2) For SEM measurements of compounds 3 and 4, which did you use, polycrystalline powders or crushed single crystals? In addition, what is the difference of electrochemical property between polycrystalline powder and crushed single crystal?

(3) Please explain why morphology is important rather than crystal structure.

Author Response

This manuscript presents syntheses, solid-state structures, and electrochemical properties of lithium 1,3,5-benzenetricarboxylate (BTC) salts. Although this is a much simplest study based on Li(I)–BTC salts, the technical works are beautiful and careful. In addition, these salts, especially compounds 3 and 4, may potentially find use in applications such as Li ion batteries. Therefore, this manuscript is recommended for publication in Polymers. I have only three minor comments as below:

(1)   In the experimental section, please report the characteristic bands of IR spectrum for compound 2, which have been studied.

Response: The characteristic bands of IR spectrum for compound 2 was added.

(2)   For SEM measurements of compounds 3 and 4, which did you use, polycrystalline powders or crushed single crystals? In addition, what is the difference of electrochemical property between polycrystalline powder and crushed single crystal?

Response: We thank the reviewer for the helpful comments. The SEM measurements of compounds 3 and 4 are as-synthesized single crystals. However, the electrochemical property was achieved by the ground powder samples. The SEM measurements section was removed for less meaningful discussion.

(3)   Please explain why morphology is important rather than crystal structure.

Response: Currently, the morphology issue has less direct evidence for electrochemical property. The “morphology is important rather than crystal structure” was removed in the conclusion.

Round 2

Reviewer 1 Report

The authors have addressed the issues carefully. The work can be accepted before two minor issues resolved: 

1C means the capacity in one hour. However, what is the norminal capacity? The authors should give the norminal capacity (mAh/g) or "1 C=** mA/g".

The author names of Ref 9(i) are wrong, which should be "Liu, H.; Wang, J.-G.; Hua, W.; Wang, J.; Nan, D.; Wei, C. Chem. Eng. J. 2018, 354, 220".

Author Response

The authors have addressed the issues carefully. The work can be accepted before two minor issues resolved:

1C means the capacity in one hour. However, what is the norminal capacity? The authors should give the norminal capacity (mAh/g) or "1 C=** mA/g".

Response: We thank the reviewer for the helpful comments on capacity. In this study, 1C = 400 mA/g. We added this description in the revised manuscript.

The author names of Ref 9(i) are wrong, which should be "Liu, H.; Wang, J.-G.; Hua, W.; Wang, J.; Nan, D.; Wei, C. Chem. Eng. J. 2018, 354, 220".

Response: We thank the reviewer for the careful reading on the reference 9i. We will correct the mistake in the revised manuscript.